# Shark teeth zinc isotope values document intrapopulation foraging differences related to ontogeny and sex

Jeremy McCormack [1,2✉], Molly Karnes [3,4], Danielle Haulsee [5,6], Dewayne Fox[7] & Sora L. Kim [3]

Trophic ecology and resource use are challenging to discern in migratory marine species, including sharks. However, effective management and conservation strategies depend on understanding these life history details. Here we investigate whether dental enameloid zinc isotope ($\delta^{66}Zn_{en}$) values can be used to infer intrapopulation differences in foraging ecology by comparing $\delta^{66}Zn_{en}$ with same-tooth collagen carbon and nitrogen ($\delta^{13}C_{coll}$, $\delta^{15}N_{coll}$) values from critically endangered sand tiger sharks (*Carcharias taurus*) from Delaware Bay (USA). We document ontogeny and sex-related isotopic differences indicating distinct diet and habitat use at the time of tooth formation. Adult females have the most distinct isotopic niche, likely feeding on higher trophic level prey in a distinct habitat. This multi-proxy approach characterises an animal's isotopic niche in greater detail than traditional isotope analysis alone and shows that $\delta^{66}Zn_{en}$ analysis can highlight intrapopulation dietary variability thereby informing conservation management and, due to good $\delta^{66}Zn_{en}$ fossil tooth preservation, palaeoecological reconstructions.

[1] Institute of Geosciences, Goethe University Frankfurt, 60438 Frankfurt am Main, Germany. [2] Department of Human Evolution, Max Planck Institute for Evolutionary Anthropology, 04103 Leipzig, Germany. [3] Department of Life and Environmental Sciences, University of California Merced, Merced, CA 95343, USA. [4] Department of Earth and Atmospheric Sciences, Indiana University, Bloomington, IN, USA. [5] Department of Biology, Stanford University, Pacific Grove, CA, USA. [6] Hubbs-Seaworld Research Institute, San Diego, CA 92109, USA. [7] Department of Agriculture and Natural Resources, Delaware State University, Dover, DE, USA. ✉email: mccormack@em.uni-frankfurt.de

The lineage of the Elasmobranchii originated in the Devonian Period[1,2], and its members have since played major roles in marine ecosystems occupying trophic positions from primary consumers to apex predators[3]. In modern marine food webs, most sharks play the roles of apex and mesopredators, and their removal is likely to result in large-scale, but still uncertain, ecological consequences[4–7]. Many shark species experience global population declines, attributed to fishing, habitat degradation, climate change and pollution[8–10]. Conservation efforts will depend on a better understanding of extant shark ecology but could also be informed by how ancient species adapted to comparable environmental and climatic pressures in the past.

Stable isotope analysis can be a powerful tool to decipher an animal's life and migration history, including dietary and habitat preferences. As such, stable isotope studies are becoming more widespread in shark ecology, especially given the difficulty of directly monitoring individuals or populations[11]. Bulk carbon and nitrogen isotope values ($\delta^{13}C$, $\delta^{15}N$) analysed in an animal's organic tissues, including bone and tooth collagen, are the most applied dietary and trophic level proxies[12–14]. Generally, both $\delta^{13}C$ and $\delta^{15}N$ values increase with trophic level due to the preferential loss of the lighter isotopes $^{12}C$ and $^{14}N$ in respiration and urea, respectively[15,16]. However, $\delta^{13}C$ values behave more conservatively, increasing ~1‰, or not at all, per trophic level compared to $\delta^{15}N$ values, which increase on average by 3.4‰ per trophic level in most tissues[17–19]. Thus, $\delta^{13}C$ values are more commonly applied to identify the primary producer(s) in a food web[20,21], whereas $\delta^{15}N$ values are used to estimate trophic levels.

The application of bulk $\delta^{15}N$ values as a trophic level proxy relies on the assumption of predictable, constant nitrogen isotopic fractionation factors between the diet and the examined consumer tissue. Yet, physiological and environmental factors can also influence diet-tissue fractionation factors shifting them from their predicted values[22–26]. In addition, the traditional bulk collagen $\delta^{13}C$ and $\delta^{15}N$ method used in an archaeological and palaeontological context is not applicable to 'old' fossil material (>100,000 years) as it is limited by the degree of protein preservation[27]. In recent years, other non-traditional dietary/trophic isotopic proxies have emerged that can complement the traditional $\delta^{15}N$ proxy to provide additional otherwise inaccessible information, as well as act as a substitute in ancient sample material with poor or no collagen preservation[28–34]. Among these methods, zinc isotope ratios ($^{66}Zn/^{64}Zn$ expressed as $\delta^{66}Zn$) were recognised as a particularly promising trophic level proxy in mammals[29,35] and fish[33]. Pristine, i.e., diet-controlled $\delta^{66}Zn$ values can be preserved in shark tooth enameloid for millions of years, making it an effective tool to investigate deep-time shark ecology and evolution[33].

Zinc is supplied into vertebrate tissues through diet[36,37] and undergoes mass-dependent fractionation within an organism[38–40]. Zinc intra-organism isotopic fractionation is tissue-specific, depending on the Zn coordination environment[38–40]. Heavier Zn is typically concentrated in stiffer bonds, i.e. heavier Zn preferentially binds to ligands with a stronger electronegativity (oxygen>nitrogen>sulphur)[38,39,41]. This preference results in a depletion of $^{66}Zn$ in organs and muscles relative to the animal's diet leading to successively lower $\delta^{66}Zn$ values as trophic levels increase. For ecological, archaeological, and palaeontological studies, $\delta^{66}Zn$ is commonly measured in the mineral phase of skeletal tissues (bioapatite) of bones and teeth[29–33,35]. Notably, there appears to be a predictable offset in $\delta^{66}Zn$ values when analysing different bioapatite tissues of the same individual, with bone/tooth dentine having values that are on average 0.2‰ higher compared to tooth enamel/enameloid[33,35]. Per trophic level, bioapatite $\delta^{66}Zn$ values decrease by approximately 0.4‰ in both marine and terrestrial

mammals, although the exact trophic level fractionation factors have yet to be determined experimentally[32,35].

Dissolved zinc is highly depleted in the surface ocean with typical concentrations of 0.01–0.5 nmol kg⁻¹, while deep water usually exhibits higher concentrations, e.g. up to 10 nmol kg⁻¹ in the North Pacific[42]. Despite this nutrient-like Zn distribution, which is a result of biological uptake, there is no consistent covariation between dissolved Zn concentration and isotopic composition[42]. The shallow upper ocean exhibits a large $\delta^{66}Zn$ variation for dissolved Zn ranging from −1.1 to +0.9‰[43,44]. This variability and its causes are still the subject of debate (reviewed by refs. [42,45]). In contrast, dissolved $\delta^{66}Zn$ values from deep water are near homogenous at approximately +0.45‰[42,43,46,47]. Marine Zn sources, mainly stemming from rivers and aeolian dust, are also relatively homogenous with $\delta^{66}Zn$ values around +0.33‰[48], close to the average upper continental crust value of +0.3‰[45,49].

Few studies have investigated zinc isotope variability in marine vertebrates[29,32,33]. These initial results suggest that, within the same tissues analysed (e.g. bone, enameloid), zinc isotope values have a similar range across geographic locations within the same taxa/trophic levels, implying little variation in marine Zn isotope food web baselines, perhaps even on geological timescales[33]. However, large-scale marine food web baseline data derived from primary producers are currently non-existent, which limits any detailed discussions of marine Zn isotope baselines at this time.

Notably, $\delta^{66}Zn$ and $\delta^{15}N$ are measured in different components of skeletal tissues, bioapatite and collagen respectively, and are metabolically and environmentally independent of each other. As such, a combined analysis of both trophic level proxies offers the potential of gaining additional ecological information, cross-verification of results and correction for environmental and/or physiological factors influencing either proxy[50]. The inclusion of $\delta^{66}Zn$ may also provide a more direct comparability of marine species trophic ecology between spatially and temporally distinct locations than possible by traditional isotope analysis alone. While it remains unknown for $\delta^{66}Zn$ values, both bulk $\delta^{13}C$ and $\delta^{15}N$ values are known to be strongly affected by marine baseline variability[26,51,52].

Here, we test, whether enameloid $\delta^{66}Zn$ ($\delta^{66}Zn_{en}$) values can be used to investigate foraging differences within a single marine predator population. We analyse tooth dentine collagen $\delta^{13}C$ and $\delta^{15}N$ ($\delta^{13}C_{coll}$, $\delta^{15}N_{coll}$) values with $\delta^{66}Zn_{en}$ values of the same tooth from 53 sand tiger shark (Carcharias taurus) individuals that were caught and released in Delaware Bay (New Jersey, USA). Declines in abundance are observed globally for C. taurus[53]. This species is particularly vulnerable to over-exploitation due to slow growth, late maturation, and low reproductive output[54,55]. Subsequently, C. taurus is now listed as "Critically Endangered" on the International Union for Conservation of Nature (IUCN) Red List of Threatened Species[56]. Understanding this species feeding and habitat requirements is essential to facilitate conservation and management efforts.

Carcharias taurus is a large coastal shark that primarily feeds on teleosts and smaller elasmobranchs[57,58]. This species is the ideal subject for a study such as this; Carcharias taurus is the only extant member of a genus who's now extinct members, along with analogue Odontaspididae (e.g. Striatolamia), were globally distributed and abundant throughout the Cenozoic, dating back to the Cretaceous period[59–61]. Carcharias is thus a key genus for palaeontological and conservation palaeobiological studies. In addition, while C. taurus is wide-ranging in its distribution, extant populations are isolated and require conservation efforts that are tailored to the population-specific regional ecology[62]. Carcharias taurus is a migratory species and previous studies suggested differences in the timing and routes of migration between juvenile and adult individuals as well as males versus

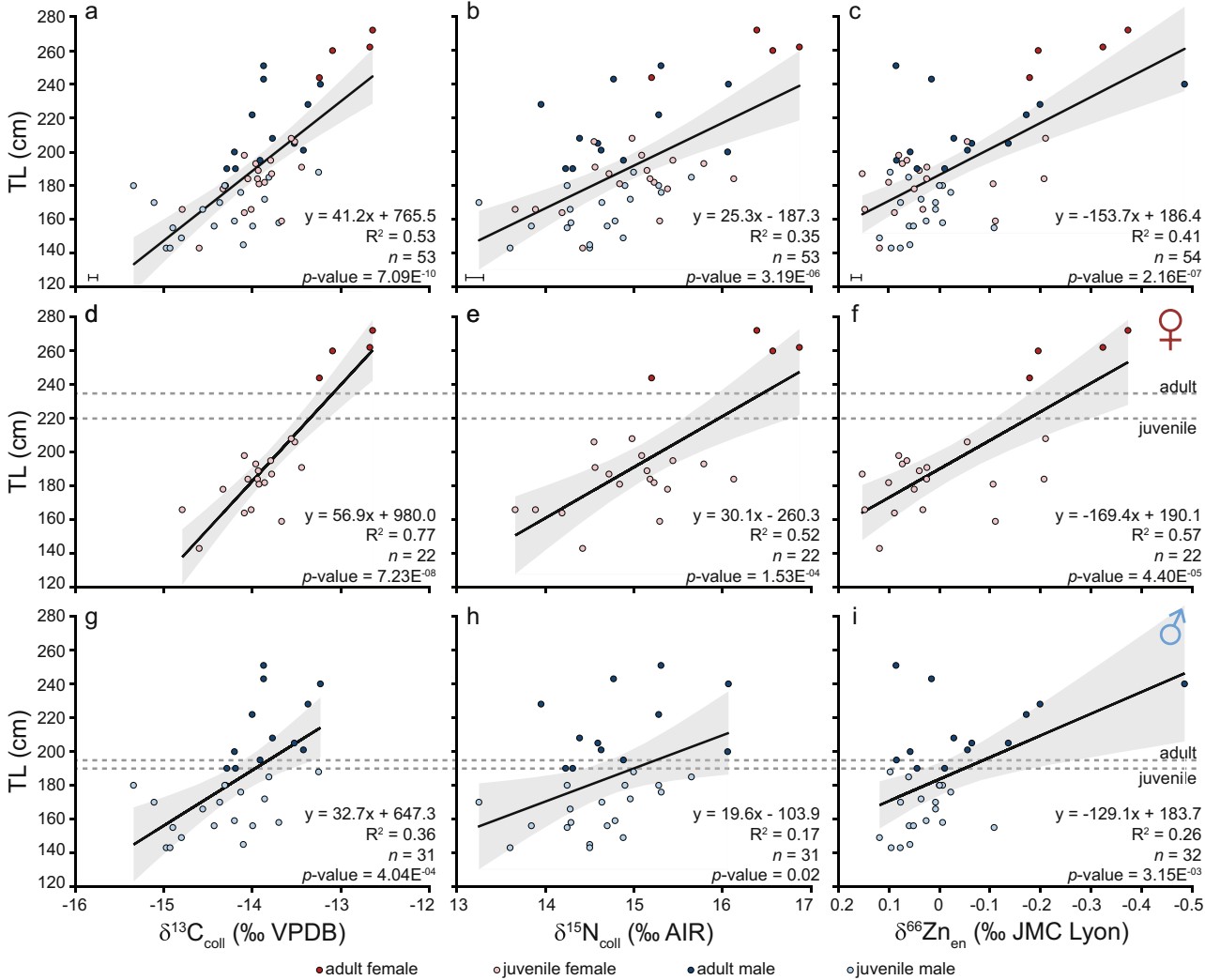

**Fig. 1 Biplots between *Carcharias taurus* total length (TL) in cm and $\delta^{13}C_{coll}$, $\delta^{15}N_{coll}$ and $\delta^{66}Zn_{en}$ values.** All individuals are depicted in **a–c**. Only female individuals are depicted in panels **d–f** and only males in panels **g–i**. The dotted lines indicate TL at maturity following refs. [67,68] with darker coloured symbols indicating individuals considered to be adults in this study. Trend lines show the linear regression fit and grey shaded areas show 95% confidence intervals for all data points shown in the respective panel. Error bars represent analytical uncertainty (1 SD) derived from standard replicate analyses and $n$ indicates the number of *C. taurus* individuals. Zinc isotopes are depicted on an inverted x-axis for readability (**c, f, i**).

females within the western Atlantic population[55,63]. Yet, little is known about migration, sex, and size-related differences in their foraging ecology. This multi-proxy isotope study investigates the isotopic variability and niches within this population and associated differences in resource use related to sex, ontogeny and migration patterns. Finally, we discuss the implications of our $\delta^{66}Zn_{en}$ results for ecological and palaeobiological studies investigating intrapopulation dietary variability.

## Results and discussion
Modern ecological studies often rely exclusively on $\delta^{15}N$ values from organic tissues to establish trophic levels in marine predators, especially as foraging ecology typically cannot be directly observed. Despite its merits, metabolic and environmental factors may compromise $\delta^{15}N$ trophic level interpretations[22–26]. In addition, organic substances are commonly not available for fossil studies. Here, we measure $\delta^{66}Zn_{en}$, an emerging trophic level indicator preservable in mineralised tissues over geological timescales[33], together with bulk dental $\delta^{13}C_{coll}$ and $\delta^{15}N_{coll}$ values in western Atlantic *C. taurus* individuals. All three dietary indicators vary substantially with significant correlations with body

size, and, thereby, ontogeny (Fig. 1a–c). The heterogeneous $\delta^{66}Zn_{en}$ values and their correspondence to body size, and $\delta^{13}C_{coll}$ and $\delta^{15}N_{coll}$ values (Fig. 2g, h) indicates that the diet variability of this *C. taurus* population is well represented by its $\delta^{66}Zn_{en}$ values. These results demonstrate that zinc isotopes record diet with high fidelity and serve as a trophic level indicator even on an intrapopulation level.

**Isotope compositions of *Carcharias taurus* teeth.** There is a large isotopic variability among the teeth of different individuals for all three isotopes measured. Collagen $\delta^{13}C$ values have a range of 2.7‰, from −15.3 to −12.6‰, with a mean of −14.0 ± 0.6‰ ($n = 53$). Collagen $\delta^{15}N$ values have a range of 3.6‰, from +13.3 to +16.9‰, with a mean of +14.9 ± 0.8‰ ($n = 53$). Enameloid $\delta^{66}Zn$ values have a range of 0.64 ‰, from −0.49 to +0.15‰, with a mean of −0.02 ± 0.13‰ ($n = 54$). All three isotopes show a significant variation in body size, with relationships decreasing from $\delta^{13}C_{coll}$ to $\delta^{66}Zn_{en}$ to $\delta^{15}N_{coll}$ (Fig. 1a–c). Physiological effects that could affect all three isotope systems for all studied individuals, leading to a correlation with body size, may be related to growth rate or body size-related differences in isotopic

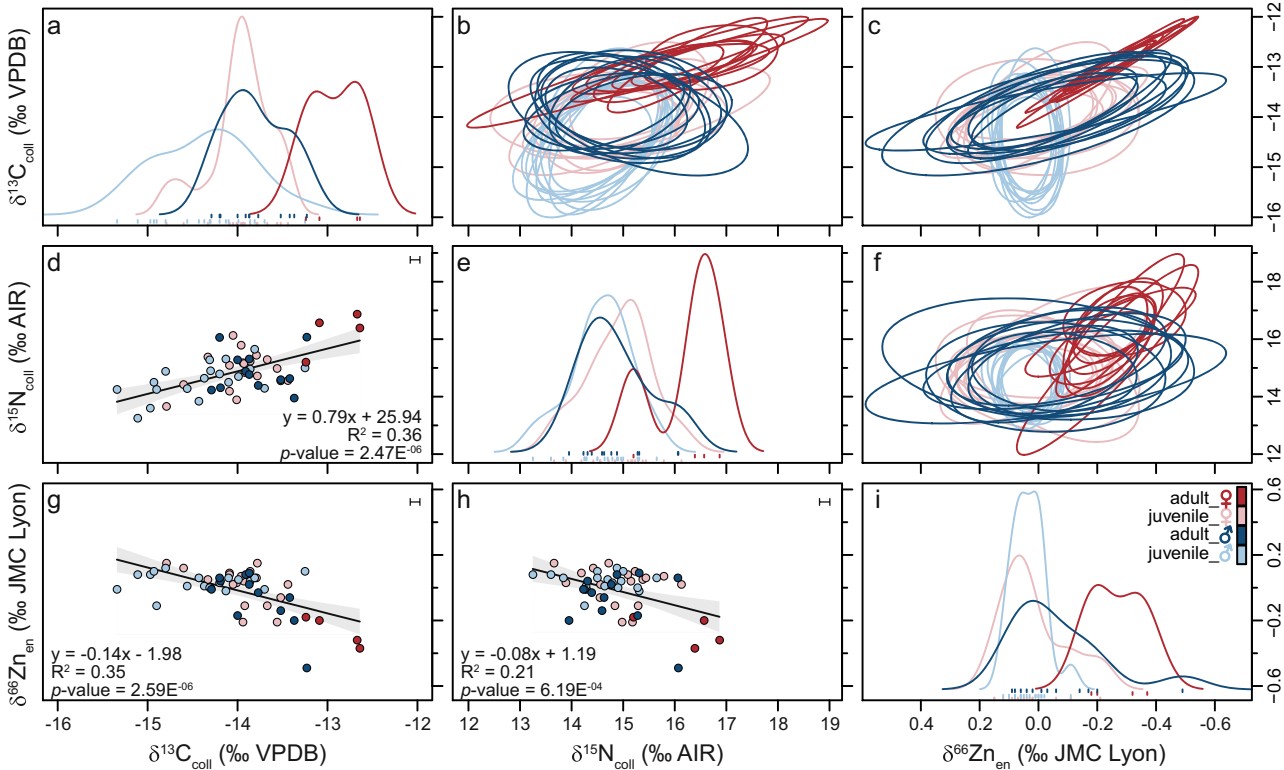

**Fig. 2 Isotopic niche plots, density distributions and scatter plot data for *Carcharias taurus* individuals.** Isotopic niche plots are depicted in **b**, **c** and **f**, density distributions in **a**, **e** and **i**, and scatter plots in **d**, **g** and **h**. Scatter plots include linear regression fits with 95% confidence intervals (grey shaded area) for all data points shown in the respective panel (in all cases, the number of analysed individuals is $n = 53$). *Carcharias taurus* individuals are separated by sex (female/male) and total length (juvenile/adult). Plots were produced using the R package nicheROVER[97]. The isotopic niche region is projected as ten random ellipses for each *C. taurus* group and isotope pair. Error bars in **d**, **g**, **h** represent analytical uncertainty (1 SD) derived from standard replicate analyses.

**Table 1 Mean $\delta^{13}C_{coll}$, $\delta^{15}N_{coll}$ and $\delta^{66}Zn_{en}$ values with 1 SD by life stage and sex for the Delaware Bay *Carcharias taurus* population.**

| Sex and ontogenetic stage | $\delta^{13}C_{coll}$ (‰ VPDB) | | $\delta^{15}N_{coll}$ (‰ AIR) | | $\delta^{66}Zn_{en}$ (‰ JMC-Lyon) | | TL range (cm) | n |
|---|---|---|---|---|---|---|---|---|
| | mean | 1 SD | mean | 1 SD | mean | 1 SD | | |
| Juvenile female | −14.0 | 0.4 | +14.9 | 0.7 | +0.02 | 0.11 | 143–208 | 18 |
| Juvenile male | −14.4 | 0.6 | +14.6 | 0.6 | +0.03 | 0.05 | 143–188 | 19 |
| Adult female | −12.9 | 0.3 | +16.3 | 0.8 | −0.27 | 0.10 | 244–272 | 4 |
| Adult male | −13.8 | 0.4 | +14.9 | 0.7 | −0.07 | 0.16 | 190–251 | 12* |

The number of individuals measured (n) for all three isotopes are given, but there was one additional adult male measured for $\delta^{66}Zn_{en}$ (i.e. $n = 13$)*. The total length (TL) range for each group is also given. Adults were identified following refs. [67,68].

fractionation factors. Other physiological effects could only impact individuals or certain groups within the population (e.g. gestating females). Differences in body size may lead to lower $\delta^{15}N$ trophic fractionation factors in larger individuals compared to smaller ones[64], but are unlikely to be as pronounced as our results, and while growth rate may impact $\delta^{15}N$ values for certain tissues, $\delta^{13}C$ values should not be affected[65]. In addition, neither of these physiological effects appear to influence $\delta^{66}Zn$ values[30,40,66]. Rather than physiological effects, the correlation of all three isotopes with body size within the western Atlantic *C. taurus* population must reflect ontogenetic niche shifts related to changes in diet composition and trophic level and/or foraging habitat, and associated variation in isotopic food web baselines.

There are noticeable differences in this population's isotopic composition between juvenile females, juvenile males, adult

females, and adult males. The four individuals, labelled as adult females, based on their size[67,68], are most distinct in their mean $\delta^{13}C_{coll}$, $\delta^{15}N_{coll}$ and $\delta^{66}Zn_{en}$ values (Figs. 1, 2, Table 1 and Supplementary Table 1). All three isotopes have a relationship with total length in females but this pattern varies for males (Fig. 1). Similarly, $\delta^{13}C_{coll}$ values show a better relationship with $\delta^{15}N_{coll}$ and $\delta^{66}Zn_{en}$ in females than in males (Fig. 3). Enameloid $\delta^{66}Zn$ values show a significant relationship with collagen $\delta^{15}N$ values in females but not in males (Fig. 3). Isotopic niche regions (N$_R$) differ for all four *C. taurus* groups (i.e. juvenile females, juvenile males, adult females, adult males, Fig. 2). Adult females have the lowest probability to be found in the isotopic niche of other groups (in all cases <20%, Fig. 4). Juvenile females and adult males have the highest probability to be found in each other's isotopic niche region (Fig. 4).

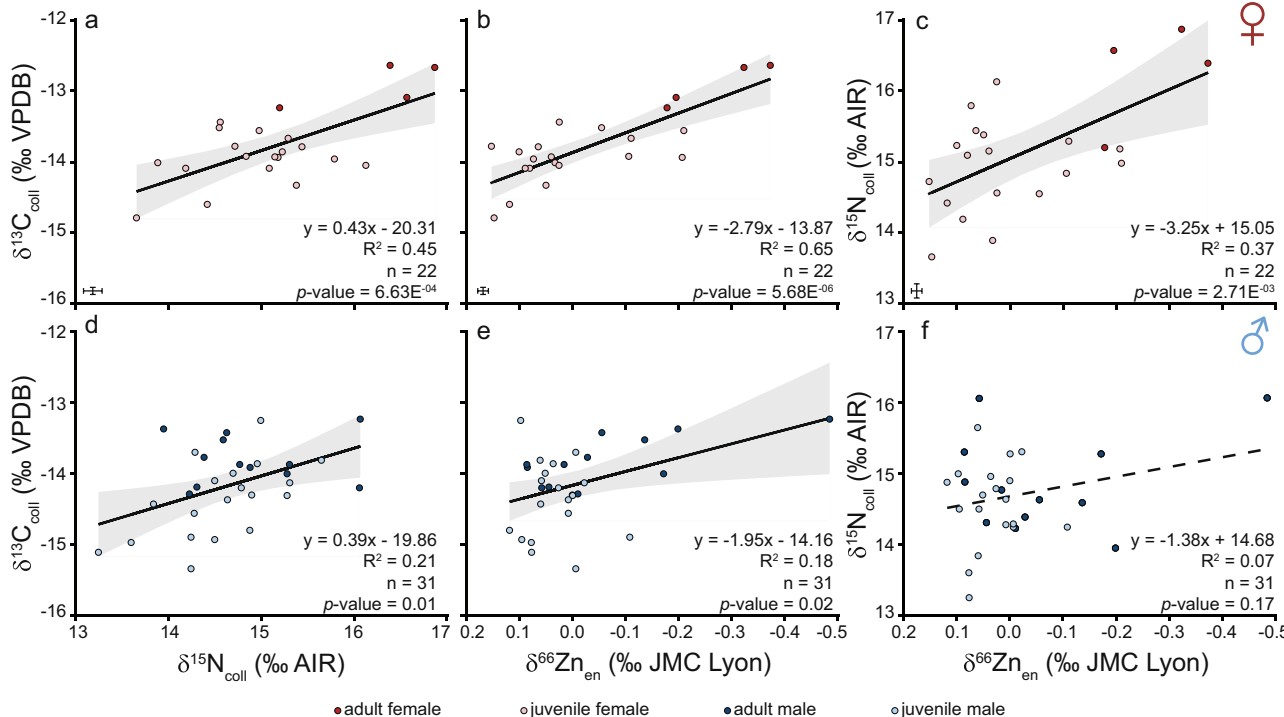

**Fig. 3 Biplots of *Carcharias taurus* $\delta^{13}C_{coll}$, $\delta^{15}N_{coll}$ and $\delta^{66}Zn_{en}$ values.** Only female individuals are depicted in panels **a**–**c** and males in panels **d**, **e**. Darker coloured symbols indicate individuals considered to be adults in this study based on their total length following refs. [67,68]. Trend lines show the linear regression fit (solid for *p* values <0.05, dashed for >0.05) and grey shaded areas show 95% confidence intervals for all data points shown in the respective panel. Error bars represent analytical uncertainty (1 SD) derived from standard replicate analyses, and *n* indicates the number of *C. taurus* individuals. Zinc isotopes are depicted on an inverted x-axis for readability (**b**, **c**, **e**, **f**).

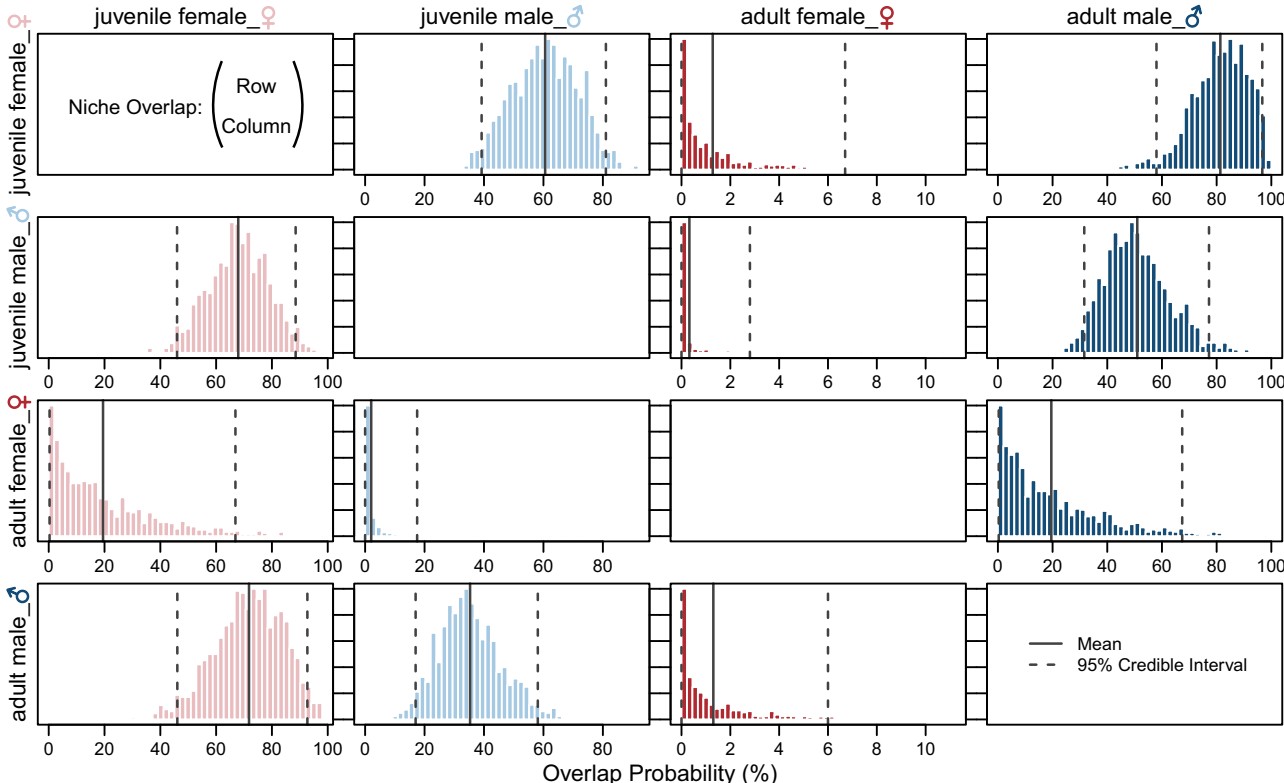

**Fig. 4 Posterior probability for individual *Carcharias taurus* to occupy the isotopic niche of another sex and ontogeny group.** Grouping and colour code are the same as displayed in Figs. 1–3. Overlap probability (%) is given for an isotopic niche region of 95%, indicating the probability of an individual from groups displayed in the rows overlapping with groups in the columns. Posterior means and 95% credible intervals are displayed as bold and dashed lines, respectively.

**Chronology of tooth formation and incorporated isotopic signals**. We document sex and ontogeny-related intraspecific $\delta^{13}C_{coll}$, $\delta^{15}N_{coll}$ and $\delta^{66}Zn_{en}$ isotope differences within the *C. taurus* population (Figs. 1, 2), reflecting a variable foraging ecology at the time of tooth formation. As the North Atlantic *C. taurus* population undertakes large-scale sex and size-segregated seasonal migrations[55,63], accurate interpretations of tooth isotopic compositions depend on reconstructing the timing of tooth mineralisation.

The *C. taurus* population examined here only spends a short time in Delaware Bay (June–September, October)[55]; therefore, the chemical composition of their teeth does not reflect the dietary uptake in late summer Delaware Bay[61]. Sharks continuously grow their teeth in a conveyor belt system and the timing of tooth formation from below the epithelial tissue to the outermost, functional position was previously quantified as ~260 days for captive Leopard Sharks *Triakis semifasciata*[13]. Further, this captive study determined the incorporation rate of dietary nutrients into dentine collagen is 32–83 days[13]. While the *T. semifasciata* has different life history traits and evolutionary history from *C. taurus*, this captive study provides an order of magnitude for the timing delay and formation rate to interpret meaningful ecological data from isotopic results. All *C. taurus* individuals and their teeth in this study were collected within a 15-day span and we expect the teeth featured here to represent a similar formation timing given each individual was only sampled once, all teeth come from the functional position, from a single population and all teeth within a row form at the same time[69]. Given the sampling date and timing for a tooth to reach the functional position via the tooth 'conveyor belt' system, we estimate a formation date (i.e. most lingual position, below the epithelium) in December 2011 (e.g. 260 days prior to August 30, 2012, is December 14, 2011). In addition, the isotopic incorporation rate of dental collagen means dietary information is integrated into organic carbon and nitrogen from September or November 2011 (e.g. 32–83 days prior to December 14, 2011). The mineralisation rate of enameloid and incorporation rate of dietary zinc is not characterised for sharks, and we expect our calculated timing of organic nutrient incorporation to be a conservative estimate.

**Diet and foraging habitat of adult males and juveniles**. *Carcharias taurus* individuals convene in Delaware Bay in the summer months, but there are variable migratory behaviours during other seasons, likely related to sex and reproductive status. Most movement observations of *C. taurus* are from acoustic tracking, which is confined to the coastal ocean and embayments where receivers are located to detect tagged animals[63,70,71]. From these studies, we know that generally, *C. taurus* departs the Delaware Bay and Mid-Atlantic region in September and October[63,71], migrates south in nearshore waters (<15 km from shore), and overwinters off the North Carolina and Florida coast[70,71]. The isotope composition from adult males and juveniles of both sexes in this study likely represents integrated dietary signals after leaving Delaware Bay on the route south from nearshore waters off the coasts of Delaware, Maryland, Virginia and North Carolina[55,63].

Adult males have the largest variability in $\delta^{66}Zn_{en}$ values spanning from +0.09 to −0.49‰ ($n = 13$). The expanded isotopic niche of adult males (and higher $\delta^{66}Zn_{en}$ variability, Fig. 2) suggests a dietary shift associated with ontogeny beyond gape-limited predation alone, which would be associated with increased trophic level. Observations from tagging data indicate that larger males dive deeper[55] where they may have access to prey unavailable to juveniles. The ability of larger-sized

individuals within a population to capture higher trophic level prey or forage in deeper depths, thereby affecting their isotopic composition, was also observed for other marine vertebrates[72]. Stomach content analyses of *C. taurus* from the Eastern Cape, South Africa revealed a broadening of dietary niche with total length; larger individuals consumed a higher taxonomic diversity that included larger prey species[58]. Similarly, South-west Atlantic *C. taurus* individuals also consume larger benthic elasmobranchs as they reach a total length >190 cm[73]. The larger isotopic niche and more variable $\delta^{66}Zn_{en}$ values suggest that adult males (Figs. 1i, 2), while opportunistic predators, are indeed able to feed upon a wider range of prey than juveniles, including from different habitats and higher trophic levels.

Adult males and juvenile females have a larger overlap of isotopic niches, whereas juvenile males differ from these two groups (Fig. 4). We hypothesise this difference is due to size since adult males are more similar in size to juvenile females in our sample set. In this study, 33% of the juvenile females and 74% of the juvenile males are <180 cm in total length. Adult males sampled were 190 to 251 cm in total length. In South Africa, a few small demersal and benthic fish were important in the diet of the smallest *C. taurus* individuals[58]. If smaller individuals are more restricted in terms of prey size, this may explain the lower $\delta^{66}Zn_{en}$ variability in juvenile males compared to adult individuals, which can feed on a wider range of prey sizes and thus trophic levels (Fig. 1i). Hence, differences in an isotopic niche between juvenile sexes may be primarily related to the size of the here examined individuals rather than sexual segregation among juveniles.

While $\delta^{66}Zn_{en}$ variability is low for juvenile males, juveniles of both sexes have a wider variability in $\delta^{13}C_{coll}$ values compared to adult individuals of the same sex, which could indicate a greater reliance of juveniles on both pelagic and demersal resources. In addition, the lowest $\delta^{13}C_{coll}$ values, <−14.5‰, are only found in juveniles of both sexes that are <180 cm in total length (Fig. 1a, d, g). This result is in line with a recent study featuring blood plasma $\delta^{13}C$ and $\delta^{15}N$ values from mainly smaller juvenile *C. taurus* from the same population[74]. In it, the authors noted that juveniles with a fork length smaller than 121 cm relied almost equally on pelagic fish and demersal omnivores (fish and squid), whereas juveniles with a larger fork length relied more heavily on demersal omnivores. In shelf environments, pelagic phytoplankton has typically lower $\delta^{13}C$ values compared to benthic primary producers[75]. Pelagic prey for juvenile *C. taurus* sampled in Great South Bay, New York, USA also had lower $\delta^{13}C$ and $\delta^{15}N$ values compared to demersal prey species[74]. Thus we tentatively interpret lower and more varied $\delta^{13}C$ values in smaller juveniles (mostly male juveniles) as a greater reliance on lower $\delta^{13}C$ value pelagic prey than in larger juveniles and adults. However, we cannot exclude the possibility that some variation may also relate to differences in migration timing with ontogeny[55], where small juveniles may forage in areas with a different baseline.

**Diet and foraging habitat of adult females**. Four individuals, which are the only adult females analysed, are distinct from the other individuals in terms of their isotopic composition for all three proxies ($\delta^{13}C_{coll}$, $\delta^{15}N_{coll}$ and $\delta^{66}Zn_{en}$) and their isotopic niche (Fig. 2 and Table 1). The likelihood of juveniles of both sexes and adult males to overlap with the isotopic niche of analysed adult females is, in all cases, less than 2% (Fig. 4). Higher $\delta^{13}C_{coll}$ and $\delta^{15}N_{coll}$ and lower $\delta^{66}Zn_{en}$ values coupled with low isotopic variability indicate that adult females generally fed at a higher trophic level and perhaps a narrower prey spectrum than juveniles and adult males. Although the number of adult female individuals is low ($n = 4$), these individuals are so distinct in terms of their isotopic composition that this trait is unlikely a

sample size artefact alone, rather indicating differences in dietary resources. Other than diet, physiological and/or environmental effects may also contribute to the observed isotopic differences.

The total length of the four adult females indicates maturity, although it is unclear whether they were gestating at the time of tooth formation. Pregnancy is one potential physiological effect that may cause intraspecific isotopic variability, especially considering adult females are most distinct from the other groups in terms of their isotopic composition for all three proxies ($\delta^{13}C_{coll}$, $\delta^{15}N_{coll}$, $\delta^{66}Zn_{en}$) and hence their isotopic niche (Figs. 2, 4 and Table 1). Previous studies focused on pregnant sharks of other species found either no differences in $\delta^{13}C$ and $\delta^{15}N$ values[76] or lower $\delta^{15}N$ values during pregnancy[77]. In other vertebrate groups, pregnancy results in lower $\delta^{13}C$ and $\delta^{15}N$ values[23,24]. Pregnancy is thus unlikely to cause the higher $\delta^{13}C_{coll}$ and $\delta^{15}N_{coll}$ values observed in the adult *C. taurus* individuals (Fig. 1). The physiological effects of pregnancy on $\delta^{66}Zn_{en}$ remain largely unexplored, but to date, sex, age and body mass have no effect on $\delta^{66}Zn$ values in terrestrial mammals, including humans[30,40,66]. While pregnancy may not impact isotopic fractionation, it could cause isotopic distinction with changes in foraging behaviour. For example, captive adult, female *C. taurus* increased their food consumption significantly during a reproductive year, which may help females prepare for investing nutrition for the large, well-developed pups of this K-selected species[78]. This physiological demand may require that adult females need to access alternative food resources than the rest of the population.

There are also behavioural changes in migration noted for adult female *C. taurus*. Pop-off satellite archival tags revealed that adult females left Delaware Bay and headed 100's km offshore to the edge of the continental shelf[55]. Species distribution models predicted that adult females were likely to occur in waters further offshore than the rest of the population in autumn[63]. It has been hypothesised that this offshore movement of adult females may be related to increased energetic demands for females in preparation for or recovery from pupping[55,63]. Adult females were also observed in North Carolina during summer[79], suggesting that not all adult females migrate annually to Delaware Bay, perhaps linked to their reproductive cycle. Pregnancy was never determined via ultrasound in female *C. taurus* individuals while residing in Delaware Bay, but there is evidence of alternate, biennial, or triennial reproductive cycles with a "resting stage" and gestational stage leading to individuals migrating north or remaining south of Cape Hatteras in summers, respectively[54,80–82]. It is possible that the isotopic distinctiveness observed in adult females (Fig. 4) is a result of different foraging habitats due to migration further offshore in autumn after leaving Delaware Bay[63] or proximity to Cape Hatteras during their reproductive cycle[79,80].

**Movement and diet inferences**. Baseline $\delta^{13}C$ and $\delta^{15}N$ values in coastal food webs are typically higher compared to those offshore[75,83,84]. For example, $\delta^{13}C$ and $\delta^{15}N$ values of particulate matter decrease along a transect from the US eastern continental shelf in the Western Atlantic to pelagic waters[85]. For *C. taurus* $\delta^{13}C_{coll}$ values, this trend seems inconsistent with the proposed foraging site of adult females further offshore east of New Jersey when compared to adult males and juveniles with lower $\delta^{13}C_{coll}$ values supposedly remaining in neritic to coastal waters. However, Northwest Atlantic $\delta^{13}C$ spatial isoscapes derived from sampling of shelf animals such as winter skates (*Leucoraja ocellata*) and loggerhead sea turtles (*Caretta caretta*) document low $\delta^{13}C$ variability along the shelf from Cape Hatters and Cape Cod[86,87]. Whereas little skate (*Leucoraja erinacea*) muscle tissue

$\delta^{13}C$ values document higher values to the northeast of Delaware Bay towards Cape Cod and lower values to the south towards Cape Hatters[87]. This $\delta^{13}C$ spatial pattern in *L. erinacea* is more consistent with the patterns observed in *C. taurus* if adult females were foraging NE of Delaware Bay and adult males and juveniles in neritic waters enroute to Cape Hatteras.

The temporal lag between tooth formation and sampling provides insights into the animal's past diet, but also complicates ecological inferences where habitat occupation is unknown. For example, if adult females foraged in waters offshore of North Carolina[79,80], regional isotopic baselines may vary compared to the foraging sites of juveniles and adult males. We are unaware of any documented significant $\delta^{13}C$ variability between the area south of Cape Hatteras and the northern neritic waters up to Delaware Bay[52,86] except for some lower values for particulate matter in neritic waters of the Eastern Shore of Virginia[85]. Due to the novelty of $\delta^{66}Zn$ as a marine trophic level proxy, baseline variability remains poorly understood, but previous studies cautiously suggest a potentially more homogenous marine baseline for $\delta^{66}Zn$ than for $\delta^{13}C$ and $\delta^{15}N$, although this remains to be tested[32,33]. Differences in $\delta^{13}C_{coll}$, $\delta^{15}N_{coll}$ and $\delta^{66}Zn_{en}$ values among sex and ontogenetic stage may also relate to a greater reliance on benthic/demersal prey by large females in lower latitude water. There is also the possibility that these four larger females consumed other elasmobranchs, which are too large to be consumed by smaller *C. taurus*[58,73,74].

**Zinc isotopes as an intrapopulation diet proxy**. The multi-isotope proxy approach combining $\delta^{13}C_{coll}$, $\delta^{15}N_{coll}$ and $\delta^{66}Zn_{en}$ values documents the distinct ontogenetic isotopic niches and sexual segregation within the Northwest Atlantic *C. taurus* population more confidently than in the case of a traditional isotope analysis alone. Despite the novelty of zinc isotope compositions as a dietary proxy, and the therewith associated interpretational uncertainties, *C. taurus* $\delta^{66}Zn_{en}$ values document ontogenetic and sexual differences in foraging ecology. Therefore, the inclusion of $\delta^{66}Zn_{en}$ values provides another independent isotope proxy to decipher intraspecific trophic level and niche variability. This is particularly important as bulk $\delta^{13}C_{coll}$ and $\delta^{15}N_{coll}$ are likely influenced to some degree also by baseline variation in distinct intraspecific habitats within the population at the time of tooth formation. Notably, $\delta^{66}Zn_{en}$ is environmentally and metabolically independent from the traditional light isotopes ($\delta^{13}C_{coll}$ and $\delta^{15}N_{coll}$), yet $\delta^{66}Zn_{en}$ values show the same distinctiveness in terms of the isotopic composition of adult females as observed for the traditional isotopes. Zinc isotope analysis of shark tooth enameloid thus allows to reconstruct not only interspecific dietary differences but also intraspecific ecological variability using both modern and fossil shark teeth[33]. Indeed, pristine dietary zinc isotope values can be preserved in fossil shark enameloid over millions of years[33]. As such $\delta^{66}Zn$ analysis can enable cross-disciplinary research with an application for modern ecological research questions, including conservation science, and palaeontology, including palaeoecology, as well as the relatively recently named field of conservation palaeobiology, aiming to increase and utilise our understanding of past ecosystems to respond to modern and future conservation challenges[88].

Uncertainties remain regarding the movement and habitat use (and therefore isotope baseline) and prey consumption of the imperilled North Atlantic *C. taurus* population. Combining $\delta^{66}Zn_{en}$ values with compound-specific nitrogen isotope analysis of amino acids from dentine collagen (and other tissues) and tagging data could further eliminate uncertainties in baseline variation and allow more detailed reconstructions of habitat resource usage of mobile predators in future studies. Such

reconstructions may be particularly valuable when targeting shark teeth, in a multi-proxy approach, across multiple rows with different tooth formation time, potentially allowing a detailed reconstruction of an individual's migration route and resource uptake along the way[14]. In any case, zinc isotope analysis is a promising tool, especially combined with other methods to decipher the resource use of extant and extinct mobile marine animals, and as such, has great potential to inform preservation efforts for endangered species, including *C. taurus*. Finally, $\delta^{66}Zn$ analysis has the potential to enable reconstructing population-specific diet variability in extinct species too, especially related to shark ontogeny, as shark tooth size scales allometrically with body size[89].

## Methods

**Sample material**. All sharks were caught legally and ethically in Delaware Bay between August 15 and 30, 2012, as part of a joint tagging programme in 2007–2015 between the University of Delaware and Delaware State University with approval from the Delaware Department of Natural Resources and Environmental Control (DNREC; 2012-021 F), Delaware State University Institutional Animal Care and Use Committee (IACUC) and the University of Delaware IACUC (1259-2014-0). After sharks were caught, biological data (i.e. sex, fork length, total length and clasper measurements) were recorded, dart tags inserted, and 1–2 teeth in the functional position were extracted (i.e. representing the oldest teeth in the jaw) before sharks were released again. Teeth that were loose were prioritised for sampling. We analysed teeth from 22 females (18 juvenile and 4 adult) and 32 males (19 juvenile and 13 adult) ranging in total length from 143 to 272 cm.

**Dental collagen carbon and nitrogen isotope analysis**. To analyse the stable isotopic compositions of organic carbon ($\delta^{13}C$) and nitrogen ($\delta^{15}N$), dentine collagen was isolated from 53 shark teeth. Powdered dentine samples were collected using a Dremel on low speed with a 300-micron diamond-tipped bit. Samples were demineralised using chilled 0.1 M HCl, then rinsed five times with deionised water, before being freeze-dried overnight[90]. Samples were weighed to 0.4–0.5 mg in $3 \times 5$ mm tin capsules. All collagen samples were measured for $\delta^{13}C$ and $\delta^{15}N$ values using a Costech 4010 Elemental Analyser coupled to a Delta V Plus continuous flow isotope ratio mass spectrometer with a Conflo IV in the Stable Isotope Ecosystem Lab (SIELO) of the University of California Merced. All data were corrected for linearity and drift using a suite of calibrated reference materials: USGS 40 ($n = 19$), USGS 41a ($n = 10$) and costech acetanilide ($n = 10$). Long term standard deviation for the instrument is 0.1‰ for $\delta^{13}C$ and 0.2‰ for $\delta^{15}N$ values.

**Enameloid zinc isotope analysis**. Powdering of samples, zinc purification, and zinc isotope measurements were performed at the Max Planck Institute for Evolutionary Anthropology (Leipzig, Germany) following the protocol as described in ref. [33]. Zinc isotopes were measured from tooth enameloid of 54 sand tiger individuals (i.e. including one individual for which $\delta^{13}C$ and $\delta^{15}N$ values could not be analysed). All teeth were cleaned by ultrasonication in ultrapure water (Milli-Q water) for 5 min and dried in a drying chamber at 50 °C overnight. Enameloid samples were abraded from the top surface using a dental drill. The enameloid powder were dissolved in closed perfluoroalkoxy vials with 1 ml 1 M HCl on a hotplate for 1 h at 120 °C and then evaporated. The residue was then dissolved in 1 ml 1.5 M HBr and placed in an ultrasonic bath for 30 min. Zinc purification was performed in two steps, following the modified ion exchange method adapted from ref. [91], described in ref. [35], and always included a chemistry blank and reference standard (NIST SRM 1400) to monitor contamination and Zn elution. One ml of AG-1×8 resin (100–200 mesh) was placed in 10 ml hydrophobic interaction columns (Macro-Prep® Methyl HIC). The resin was then cleaned twice with 5 ml 3 % HNO₃ followed by 5 ml ultrapure water and subsequently conditioned with 3 ml 1.5 M HBr. Following sample loading, 2 ml HBr were added for matrix residue elution, after which Zn was eluted with 5 ml HNO₃. After performing the column chromatography twice, the solution was evaporated for 13 h at 100 °C and the residue re-dissolved in 1 ml 3% HNO₃ for analysis.

Zinc isotope measurements were performed using a Thermo Fisher Neptune MC-ICP-MS. Instrumental mass fractionation was corrected by Cu doping following the protocol of Maréchal et al.[92] and Toutain et al.[93]. The in-house reference material Zn Alfa Aesar-MPI was used for standard bracketing. All $\delta^{66}Zn$ values are expressed relative to the JMC-Lyon standard material (mass-dependent Alfa Aesar-MPI offset of +0.27‰ for $\delta^{66}Zn$[35,94]). Zinc concentrations in the respective samples were estimated following a protocol adapted from one used for Sr by ref. [95], applying a regression equation based on the Zn signal intensity (V) of three solutions with known Zn concentrations (150, 300 and 600 ppb). The $\delta^{66}Zn$ measurement uncertainties were estimated from standard replicate analyses and ranged between ±0.02‰ and ±0.04‰ (2 SD). Samples were typically measured at least twice with mean analytical repeatability of 0.01‰ (1 SD, $n = 53$). Reference

material NIST SRM 1400 was prepared and analysed alongside the samples and had $\delta^{66}Zn$ values (0.92 ± 0.02‰ 1 SD, $n = 6$) as reported elsewhere[32,96]. Reference materials and samples show a typical Zn mass-dependent isotopic fractionation, i.e. the absence of isobaric interferences, as the $\delta^{66}Zn$ vs. $\delta^{67}Zn$ and $\delta^{66}Zn$ vs. $\delta^{68}Zn$ values fall onto lines with slopes close to the theoretic mass fractionation values of 1.5 and 2, respectively (Supplementary Data 1).

**Statistics and reproducibility**. Sharks were sorted into four groups separated by sex (female/male) and total length (juvenile/adult). Females and males are considered adults when the total length reaches 220 and 190 cm, respectively[67,68]. The isotopic tracers $\delta^{13}C_{coll}$, $\delta^{15}N_{coll}$ and $\delta^{66}Zn_{en}$ were used to quantify the isotopic niche size and overlap among the groups within the Delaware Bay sand tiger shark population using the R package nicheROVER[97]. Using nicheROVER we estimated the isotopic niche per group in a three-dimensional ($\delta^{13}C_{coll}$, $\delta^{15}N_{coll}$ and $\delta^{66}Zn_{en}$) space. NicheROVER uses a probabilistic method to calculate our three-dimensional isotopic niche regions ($N_R$) using a Bayesian framework. Niche overlap was calculated from 1000 Monte Carlo draws with an alpha = 0.95 probability level and is defined as the probability of finding an individual from one group in the $N_R$ of another group[97].

**Reporting summary**. Further information on research design is available in the Nature Portfolio Reporting Summary linked to this article.

## Data availability
All data generated during this study are included in this published article (Supplementary Data 1).

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

## Acknowledgements

This research was funded by the Deutsche Forschungsgemeinschaft awarded to J.M. (Award Number 505905610), National Science Foundation Sedimentary Geology and Paleobiology Award to S.L.K. (Award Numbers 1830480 and 2239981) and the Max Planck Society. S. Steinbrenner and M. Trost (Department of Human Evolution, Max Planck Institute for Evolutionary Anthropology, Leipzig) and R. Trayler (Stable Isotope Ecosystem Laboratory of the University of California, Merced) are thanked for technical and analytical support. The fieldwork in which teeth samples were collected was generously supported by the Robertson's Fund, the Delaware Space Grant Consortium, and the DuPont Clear into the Future Programme Award to Delaware State University's (DSU) Center for Integrated Biological and Environmental Research, Delaware EPSCor with funds from the National Science Foundation Grant (EPS0814251), and the NOAA-NMFS Proactive Conservation Programme Award (NA09NMF4720365). We would also like to thank M. Oliver, M. Breece, J. Kilfoil, C. Simpson and G. Reger for their assistance in the field.

## Author contributions

S.L.K. and J.M. designed the study. J.M. performed the $\delta^{66}Zn_{en}$ analyses, and M.K. and S.L.K. performed the $\delta^{13}C_{coll}$ and $\delta^{15}N_{coll}$ analyses. M.K. performed the statistical analyses. J.M., M.K., D.H., D.F. and S.L.K. analysed and interpreted the data. J.M. wrote the initial manuscript draft. All authors contributed to editing the final version of the manuscript.

## Funding

## Competing interests

The authors declare no competing interests.
