## [Peer Review File · Communications Biology]

Reviewers' comments:

Reviewer #1 (Remarks to the Author):

This is an interesting study, particularly notable for the use of Zn stable isotopes to compliment more traditional C and N isotope analyses for the inference of tropho-spatial ecology. The authors should be commended not only on producing the analytical data, but also for the collection of a sample suite that is well constrained in terms of sample collection.

That said, I'm afraid the results and data interpretation remain rather speculative – it is difficult to understand clearly how the case study moves us forwards significantly given the limitations imposed by small sample sizes (for mature females $n=4$) and a lack of understanding of the ecology of the case study group to base inferences from Zn isotopes on..

I think there is a very interesting story in here – the fact that all three isotopes show some co variance with body size is clearly meaningful, and I would be intrigued to see more in depth discussion of why that might be. However, the common covariance with body size confounds some of the analyses of niche area and ecological inference.

The bulk of the discussion is rather speculative drawing possible causes of observed isotopic variation based on what might be expected for this (or other) species... The discussion is perhaps more suited to a more marine ecology focussed journal.

Overall, then I think there is interesting data here and a well-executed analysis which is clearly written up and explained – but given the remaining uncertainties in the inferences that can be drawn from the data I'm not convinced that the methodological / conceptual advance is enough to engage a broad cross-disciplinary audience.

Abstract

To me, it seems like the abstract is slightly oddly focussed - the original and novel aspect of this work is the addition of Zn isotopes to traditional 2 (or 3) isotope systems used to infer aspects of tropho-spatial ecology – but this novelty is slightly lost in the abstract. I would like to see re-phrasing to highlight the case study application of an interesting and potentially valuable ecological tracer. I think that would increase the readership interest.

Main text

Similar critique could be levelled at the first paragraph – again I'm not sure that sand tiger shark ecology, important as it might be, is especially interesting to a wide scientific audience – but the development of a new isotopic tracer with ecological value certainly is. I would lead on that with the case study application as a secondary (but valuable and interesting) component.

Line 44-50 – what is the underlying mechanism or at least, best current theories for reduction in $\delta^{66}\text{Zn}$ values with trophic level? Line 52, stating that variations in $\delta^{66}\text{Zn}$ values are metabolic[ally] independent from those in $\delta^{15}\text{N}$, $\delta^{13}\text{C}$ values – needs backing up – again with reference to best interpretations of mechanisms underpinning reduction in $\delta^{66}\text{Zn}$ values with trophic level.

Line 57 – perhaps a little more exploration of potential for 'baseline' effects on $\delta^{66}\text{Zn}$ is needed?

Results – here I feel the results need more data analysis. You are largely exploring the correlated

variation of the three isotope systems with body size - with potential interacting effects of sex and age group.. Given body size has a positive (if weak in places) co-variance with each of the isotope systems -there is inter-dependence. Does this co-variance with body size reflect trophic level effects on all isotope systems? While the slopes of regression lines are not given in the main text, this seems unlikely as slopes for TL vs $\delta^{13}\text{C}$ values appear relatively high - but it seems important to discuss the underpinning reasons for observed (co) variance in the three isotopes analysed - with a particular focus on Zn...

The common effect of body length creates issues for the niche size analyses, as niche size is therefore influenced by the body size range and distribution within each group - so unless that is standardised somehow, the niche parameters are partially descriptions of body size (and sample size)...

I do not think it is reasonable to draw ecological inference from a reduction in apparent isotopic niche sizes when the population sizes are $n=18$ and $n=4$. Clearly $n=4$ is insufficient to infer among-individual variability....

There is a lot of speculation about potential ecological 'meaning' of observed isotope values in the discussion. Given limited information on foraging ecology for this species and relatively low sample sizes when broken down to sex / age categories- is it perhaps the best case study to try to demonstrate ecological value of Zn isotopes in the context of shark ecology?

Figures - no confidence interval on regression slopes? Can we have the slopes reported?

Re ethics approval - For my institution I would need to obtain ethics approval for re-purposing samples taken in previous field campaigns - especially if the sampling did not have specific approval for removal of live animals or manipulation of animals. Generally making use of existing biological material is seen as good practice, but not free from ethical responsibility - so I would perhaps refine a statement that ethical approval is not necessary. Some oversight is probably good practice.

Reviewer #2 (Remarks to the Author):

The manuscript is interesting. This is the first time I see the use of the Zinc isotope as a reference in trophic information and even related to the trophic level. This approach and even more so in sharks is quite useful specifically because it allows us to corroborate what is found in the ^{15}N isotopes and relate it to the environmental part in terms of the conditions of the environment where predators tend to feed. I suggested some changes in the title because it refers only to the Zinc isotope, and they work with N and C as well. Additionally, there are some confusing sections that are worth reviewing. I consider that much of the discussion is well addressed although some sections related to mature females justify that the isotope values could be related to reproductive processes, but at no point do they mention that the mature females had any sign that they were gravid females, so this should be handled with caution. However, I consider that the manuscript is very interesting and will allow knowing how these new isotopes have a utility in eco-trophic studies specifically in marine predators.

Specific Comments

- Title: Why if the manuscript is focused on trophic ecology using SIA tools of ^{13}C , ^{15}N just mentioned Zinc... I would like to suggest modifying the title... could be: "Shark teeth isotope values document intrapopulation foraging differences related to ontogeny and sex"
- Lines 40-41. The ^{13}C does not always increase with the trophic level at least not in top predators such as sharks. Several species had shown that does not exist a positive relationship between ^{15}N and ^{13}C ... is logical to think that ^{15}N must increase with the trophic level is the way to interpret this

isotope. But the ^{13}C most of the time showed an opposite trend or any relationship.

- Line 82 Figure 1. The panel d-f represents the data from Males... why only put panels d and e... please check this.
- Lines 139-141. How could be possible that the information that you got from the teeth of sharks caught in August of 2012 reflects the diet and habitat condition from October to December 2011? I know that does not exist the turnover rate for this species but you previously mentioned that considering the captivity study from *T. semifasciata* was ~ 260 days and the incorporation rate of dietary nutrients into dentine collagen is 32 – 83 days. The time that is proposed that reflects these isotopes even is more than 260 days (close to 8.6 months).. Please you need to be more detailed in this approach because is confusing.
- Lines 170-172. The $\delta^{66}\text{Zn}$ start to decrease with the trophic level is logical though the mature individual has the lowest value, so Why you mentioned that the difference in juvenile males could be related to the less variety of prey level? I agreed that juvenile males could consume prey with lower compared with adult males... these lines are a little confusing.
- Lines 209-2015. The 4 mature females were pregnant?. In the document, I didn't find anything related to this. These lines aboard the importance of this process could affect more the ^{66}Zn in comparison with the other isotopes. but what's the point of this?

Author response

We are thankful for the very thorough and constructive feedback given by the two reviewers. Implementing the reviewers' comments has significantly strengthened our arguments. Specifically, we now address the novelty and the cross-disciplinary aspects of our research much more clearly in the abstract, introduction and discussion. We also document and discuss much more clearly the underpinning reasons for the covariance of all three isotope systems with body size. We now plot the covariance of all three isotopes with body size (with all data undifferentiated by sex) to clearly show the correlation of all isotopes with body size independent of sex and the adult female samples size issue (Figure 1 a – c). We then discuss why this correlation reflects differences in ontogenetic niche (diet/trophic level and/or habitat), before going into a more detailed discussion. This way we more clearly show that Zn isotopes can be used to trace ontogenetic niche shifts (diet and/or habitat) which can be applied to various disciplines, including conservation biology, archaeology, palaeontology, ecotoxicology, fisheries sciences and more.

We have also toned down and shortened our discussion regarding the adult female ecology, thereby focusing now more on the general aspects of how Zn isotopes can be used to infer intrapopulation dietary differences alone and in combination with other proxies.

In the following we discuss the comments made by each referee (original remarks are in black, our rebuttal/comments in blue). We also numbered the reviewer comments, allowing us to refer to comments made by each reviewer and our responses. When referring to changes made with line numbers, we refer to the tracked changes version of the manuscript unless otherwise specified.

Reviewers' comments:

Reviewer #1 (Remarks to the Author):

1.1 This is an interesting study, particularly notable for the use of Zn stable isotopes to compliment more traditional C and N isotope analyses for the inference of tropho-spatial ecology. The authors should be commended not only on producing the analytical data, but also for the collection of a sample suite that is well constrained in terms of sample collection.

We thank the reviewer for his appreciation of our work.

1.2 That said, I'm afraid the results and data interpretation remain rather speculative – it is difficult to understand clearly how the case study moves us forwards significantly given the limitations imposed by small sample sizes (for mature females n=4) and a lack of understanding of the ecology of the case study group to base inferences from Zn isotopes on..

We respectfully disagree with the reviewer here. Interpretations of proxy data must be by definition somewhat speculative, as proxy data is a stand in for direct measurements. Especially in marine isotope geochemistry different effects that might impact the data must be discussed, as dietary stable isotope data can be influenced not only by dietary, but also habitat and environmental, and physiological factors. Ignoring different potential effects would not be good scientific practice, which is why we discuss them in the manuscript. Nevertheless, our data and interpretation clearly documents dietary differences related to ontogeny within the population demonstrated by three independent proxies, including the novel Zn isotopes in enameloid, which is clearly a significant advancement for marine ecology and by extension palaeobiology and more related disciplines. We now address the interdisciplinary scope much more clearly in the manuscript. We also added to

Figure 1 a plot showing the covariance of all three isotopes with body size for all data independent of sex (a – c). We thereby explore our data more adequately demonstrating the ontogenetic niche variability throughout the entire population before going into the discussion about the individual population groups.

We understand the reviewers point about the low sample size for adult females (see also our response to comment 1.12, 1.13 below). While it is unfortunate that only four adult females were available for this study (and we do not have access to more at this time), the clear distinctiveness of their isotopic signals documents without a doubt that the ecology of these four individuals clearly differed from all other individuals sampled here. We already noted that in the previous manuscript version. Naturally we cannot say if all females show this distinct isotopic signal, but the fact that only the largest female individuals, which based on their size reached maturity, show this signal is clearly worth discussing in the manuscript. In addition, the fact that adult females were encountered much less frequently than adult males and juveniles (hence the low sample size) only supports the literature (which we cite) and our isotopic interpretation, i.e., not all females visit Delaware Bay annually and that these four females were likely foraging in a different habitat and/or prey than all other individuals at the time of tooth formation. Hence the difference in their isotopic signal for three independent dietary tracers, which we cannot explain otherwise. Nevertheless, we have toned down and shortened our discussion on the adult females.

1.3 I think there is a very interesting story in here – the fact that all three isotopes show some covariance with body size is clearly meaningful, and I would be intrigued to see more in depth discussion of why that might be. However, the common covariance with body size confounds some of the analyses of niche area and ecological inference.

We now added to Figure 1 a plot showing the covariance of all three isotopes with body size for all data independent of sex (a – c) and discuss in more detail the potential factors for these correlations. We demonstrate in the discussion more clearly that this covariance must relate to ontogenetic niche shifts, i.e., these correlations must relate to changes in diet/trophic level and/or habitat. We do this by excluding the possibility of physiological effects influencing all three dietary proxies in a similar fashion across all studied individuals and by extension the entire population (lines 137-147, 153-164). Only then do we discuss differences among body sizes and sex in more detail in relation to likely causes thereof related to body size differences in prey availability and habitat differences at the time of tooth formation based on known and hypothesised migration and dietary behaviour.

1.4 The bulk of the discussion is rather speculative drawing possible causes of observed isotopic variation based on what might be expected for this (or other) species... The discussion is perhaps more suited to a more marine ecology focussed journal.

Again, we now address in more detail the overall correlation with all three proxies with body size (see also responses to comments 1.2, 1.5, 1.6, 1.7), added to Figure 1, shortened and toned down the discussion on the adult females, and expanded on the implications for multidisciplinary research in the abstract, introduction and final discussion chapter “Zinc isotopes as an intrapopulation diet proxy”. Thereby we now more clearly focus on the implications using this novel Zn isotope proxy in (palaeo)ecological studies and beyond, unambiguously highlighting why our results are evidently of importance to biologists and researchers in comparable disciplines, regardless of sub-discipline.

1.5 Overall, then I think there is interesting data here and a well-executed analysis which is clearly written up and explained – but given the remaining uncertainties in the inferences that can be drawn

from the data I'm not convinced that the methodological / conceptual advance is enough to engage a broad cross-disciplinary audience.

We thank the reviewer for his appreciation of our data, analysis and writing. Yet, we respectfully disagree with the reviewer regarding cross-disciplinary engagement of our research. However, we believe the reviewer's opinion here may also come from the fact that we did not highlight the most significant aspects of our results enough (the use of a new isotope system and its future applicability). This may have unintentionally underplayed the value of our contribution in terms of cross-disciplinary reach and impact especially in our abstract and introduction. This is also clear from the following two comments of reviewer 1 below, in which the reviewer suggests highlighting the use of $\delta^{66}\text{Zn}$ analysis to pique the interest of a broader readership. We have done so now (see also our responses 1.2, 1.4 and below 1.6 and 1.7).

It was always our intention to use the sand tiger sharks here as a case study to demonstrate the usefulness of this novel dietary proxy to modern ecology, conservation management, palaeobiology and beyond. Naturally, our use of this proxy within a single population may also inspire research in disciplines where dietary/trophic variability within a population has greater implications, including archaeology, anthropology, ecotoxicology or fisheries sciences. We show for the first time that this new method is sensitive enough to also trace dietary changes within a single population, not just to reconstruct trophic levels among taxa. The implications hereof for modern ecology to use as an additional tracer to cross-verify with traditional C and N isotope analysis alone merits the publication in a more general biological journal. But the application of $\delta^{66}\text{Zn}$ analysis obviously goes way beyond that, as $\delta^{66}\text{Zn}$ values are preserved in fossil shark enameloid over millions of years (McCormack et al., 2022). Therefore, our study with the sand tiger sharks, while obviously important for sand tiger ecology and conservation, may also be used as a template for other species in conservation biology, and to use in palaeobiology to investigate the diet and by implication evolution of extinct species, their life history, and changes in prey and habitat use.

In our original submission we had added a final discussion topic dedicated to the implications of "Zinc isotopes as an intrapopulation diet proxy". However, we agree, that many of the wider implications were not discussed enough, such as the use of $\delta^{66}\text{Zn}$ as a palaeontological proxy which can also be applied now to investigate dietary changes within a species/population, thanks to our ground truthing study. The implications for palaeoecological and evolutionary studies were not clearly enough implied. Again, this has changed now. We expanded on the usefulness of such a novel trophic level proxy to various disciplines in the abstract, introduction and discussion.

Abstract

1.6 To me, it seems like the abstract is slightly oddly focussed - the original and novel aspect of this work is the addition of Zn isotopes to traditional 2 (or 3) isotope systems used to infer aspects of tropho-spatial ecology – but this novelty is slightly lost in the abstract. I would like to see re-phrasing to highlight the case study application of an interesting and potentially valuable ecological tracer. I think that would increase the readership interest.

This is true and we agree that the novelty and the reason for our submission to *Communications Biology* is our use of $\delta^{66}\text{Zn}$ analysis. Please also see our response to the comments above. We have now rephrased parts of the abstract to highlight this novelty further and in doing so state now much more clearly how valuable this research can be to a wide range of biological disciplines and beyond.

Main text

1.7 Similar critique could be levelled at the first paragraph – again I’m not sure that sand tiger shark ecology, important as it might be, is especially interesting to a wide scientific audience – but the development of a new isotopic tracer with ecological value certainly is. I would lead on that with the case study application as a secondary (but valuable and interesting) component.

We agree that the development of a new isotopic tracer is of great interdisciplinary value (see our response to the two previous reviewer comments above 1.5 and 1.6). Again, it was always our intention to use the sand tiger sharks here as a case study to demonstrate the usefulness of this novel dietary proxy to modern ecology, conservation management, palaeobiology and beyond. We believe our rephrasing of the abstract, introduction and discussion have managed to highlight the novelty of our research more and the use of the sand tiger ecology as a case study. In addition, and referring to other comments of reviewer 1, we have shortened the discussion on the adult females and included additional content to our final discussion chapter “Zinc isotopes as an intrapopulation diet proxy”.

1.8 Line 44-50 – what is the underlying mechanism or at least, best current theories for reduction in $\delta^{66}\text{Zn}$ values with trophic level? Line 52, stating that variations in $\delta^{66}\text{Zn}$ values are metabolic[ally] independent from those in $\delta^{15}\text{N}$, $\delta^{13}\text{C}$ values – needs backing up – again with reference to best interpretations of mechanisms underpinning reduction in $\delta^{66}\text{Zn}$ values with trophic level.

We agree with the reviewer that our state of the art for Zn isotopes could have been more detailed. While we had given all appropriate citations for further reading, we acknowledge that given the novelty of this proxy we should have discussed the mechanisms of Zn isotope fractionation with trophic levels in more detail in the introduction. We have changed this now and expand substantially on what is known about Zn isotope ecology in the Introduction (lines 68- 80, 87-109). We also expand on why Zn and N (and C) isotopes systems are metabolically independent. We had already included a statement on the “preferential loss of the lighter isotopes ^{12}C and ^{14}N in respiration and urea, respectively” as well as discussed the increase in $\delta^{15}\text{N}$ values with trophic level. With the added information on Zn fractionation in different tissues related to its coordination environment leading to lower values in most tissues compared to the diet, the metabolic independence of both is now more clearly stated in the manuscript.

1.9 Line 57 – perhaps a little more exploration of potential for ‘baseline’ effects on $\delta^{66}\text{Zn}$ is needed?

We see now that this sentence may have been a little misleading. Here we referred to marine baseline effects in C and N, which are known to vary substantially and can complicate the interpretation of these proxies, especially in migrating animals. Therefore, having an additional environmentally independent dietary proxy can be useful to corroborate results. We have clarified this sentence now.

Due to the novelty of $\delta^{66}\text{Zn}$ as an ecological proxy, marine $\delta^{66}\text{Zn}$ food web baseline variations are poorly understood. We agree with the reviewer that it is important to properly assess also potential baseline effects on $\delta^{66}\text{Zn}$. We did this in our original manuscript in lines 256-258 and 259-266. Yet we agree again, that the Introduction could have already provided more information on the state of the art for $\delta^{66}\text{Zn}$ baseline variability. We provide this now in lines 87-101.

In addition, we like to address this comment also here by including also some more speculative statements, we wish not to include in the manuscript in great detail due to the speculative nature,

but might interest the reviewer. Indeed, baseline effects for $\delta^{66}\text{Zn}$ are observed in the terrestrial realm (Jaouen et al., 2016; 2022; Bourgon et al., 2020, 2021). But a pilot study looking at Zn isotope values in bones of marine mammals across the Arctic demonstrated a much lower variability in $\delta^{66}\text{Zn}$ across locations than what was seen for C and N, which were both shown to vary due to differences in local baselines (McCormack et al., 2021). In addition, McCormack et al. (2022) provides an extensive $\delta^{66}\text{Zn}$ dataset of both modern and fossil shark teeth values, which demonstrate that $\delta^{66}\text{Zn}$ values within a species or a taxonomic group (with similar trophic levels) are generally directly comparable in their absolute values independent of geologic age and geographic location. This is a remarkable observation because it again indicates that $\delta^{66}\text{Zn}$ baselines may be less heterogenic in their values compared to the classic $\delta^{15}\text{N}$ tracer. We can only speculate that this may be related to the relatively homogenous $\delta^{66}\text{Zn}$ values of terrestrial input (around 0.3 ‰, Little et al., 2014) and/or the relatively homogenous values of marine deep water (around 0.5 ‰, Horner et al., 2022) which should due to the increase in Zn concentration compared to surface water reflect remineralised mean biogenic Zn, which is thus likely rather isotopically homogenous. As there are no studies on marine $\delta^{66}\text{Zn}$ baseline values yet, any Zn baseline effect remains at this point unresolvable and we thus mention these as potentially causing some of the observed variation in our Zn values, yet as $\delta^{66}\text{Zn}$ clearly changes also with trophic level we anticipate trophic level effects with body size to be the main reason for our $\delta^{66}\text{Zn}$ body size correlation and differences among groups as we discuss in our manuscript. In any case, $\delta^{66}\text{Zn}$ clearly documents ontogenetic niche shifts here and we anticipate these results to spur further research on $\delta^{66}\text{Zn}$ as a marine trophic level proxy, along with issues regarding $\delta^{66}\text{Zn}$ baselines.

1.10 Results – here I feel the results need more data analysis. You are largely exploring the correlated variation of the three isotope systems with body size - with potential interacting effects of sex and age group.. Given body size has a positive (if weak in places) co-variance with each of the isotope systems -there is inter-dependence. Does this co-variance with body size reflect trophic level effects on all isotope systems? While the slopes of regression lines are not given in the main text, this seems unlikely as slopes for TL vs $\delta^{13}\text{C}$ values appear relatively high – but it seems important to discuss the underpinning reasons for observed (co) variance in the three isotopes analysed - with a particular focus on Zn...

Done. We now discuss in more detail the correlation of all three isotopes with body size, having also plotted all individuals on TL vs isotope plots (Figure 1 a – c). We discuss why this correlation must reflect differences in ontogenetic niche (diet/trophic level and/or habitat), and not physiology, before discussing dietary and habitat usage related to sex and body size. We also report the slopes of regression lines in all relevant figures now.

1.11 The common effect of body length creates issues for the niche size analyses, as niche size is therefore influenced by the body size range and distribution within each group – so unless that is standardised somehow, the niche parameters are partially descriptions of body size (and sample size)...

Naturally, we are interested in discussing how individuals of the same population with different body sizes may occupy a different niche. If ecological niches were unaffected by body size our results would not show a correlation of isotopes with body size and separate niches for the four designated groups within the population. This diet switch is not uncommon in sharks, related to difference in prey size and habitat use of larger compared to smaller individuals. By analysing the isotopic composition of three independent proxies we aimed to better constrain the isotopic niche related to

body size and sex within the population. Niche parameters are by intention depending on body size.

1.12 I do not think it is reasonable to draw ecological inference from a reduction in apparent isotopic niche sizes when the population sizes are $n=18$ and $n=4$. Clearly $n=4$ is insufficient to infer among-individual variability....

We understand the reviewers point here. However, niche size inferences should be good for most of the studied population groups despite relatively low sample sizes per group due to the small isotopic variation within each group and the here used nicheROVER ellipse based model, which is applicable even to small sample sizes (see Rossman et al., 2016). Only the adult females may show less accurate niche size results, yet the isotopic distinctiveness of these four individuals is still significant. We mentioned the limitations regarding niche size for the adult females briefly in the previous manuscript. Nevertheless, we agree partially with the reviewer and as we did not go into a detailed discussion on niche size anyway, instead we focus on isotopic niche distinctiveness and isotopic variability within the total population and among groups, we have now excluded the niche size parts of the discussion. Indeed, we agree with the reviewer that niche size comparisons are not really relevant here for the main story. We also shortened the discussion on the adult females. When looking at the entire population, the distinctiveness of the four adult females is clearly significant and hence we still discuss the likely reasons thereof, albeit with toned down inferences and clearly shortened.

Nevertheless, we feel it would be bad scientific practice not to discuss reasons why four individuals, comprising the only individuals of a specific size and sex group, have isotopic compositions distinct for three different, independent proxies and where non-isotope research indicates likely different habitat use at the time of tooth formation (see also our response to comment 1.2).

1.13 There is a lot of speculation about potential ecological 'meaning' of observed isotope values in the discussion. Given limited information on foraging ecology for this species and relatively low sample sizes when broken down to sex / age categories— is it perhaps the best case study to try to demonstrate ecological value of Zn isotopes in the context of shark ecology?

Our answer to the reviewers question can only be a clear YES. And our data clearly demonstrates the ecological value, showing within a single population differences in ontogenetic ecological niches using this novel isotopic proxy $\delta^{66}\text{Zn}$ with applications far beyond modern ecological studies. We also added to the Introduction, why this is a good species to use in such a case study (lines 110-133).

Sand tiger sharks are relatively well studied, but it is true that there are still uncertainties regarding their foraging ecology, especially for adult females. It should be mentioned here, however, that uncertainties in foraging ecology can be observed in most extant large marine migratory predators. It would be difficult to find a large marine predator, which undergoes extensive migrations, for which foraging ecology along its migration route is clearly documented. But then again, why should one apply isotopic tracers to a species whose foraging ecology is already clearly known?

The fact that there is limited information regarding the sand tigers foraging ecology is one of the reasons why it such a perfect candidate for this study, as we clearly demonstrate that Zn isotopes, with and without the traditional C and N isotopes, can provide dietary information for this critically endangered species and thereby real world applicability. Sand tiger sharks specifically are also interesting to other research areas and in the context of other sand tiger shark populations: The long fossil record of this genus dates back to the Cretaceous Period and all extant populations of this

species are highly isolated with ecological variability among them (we expand on both these points now also in the introduction).

Some speculation on the isotopic controls in the discussion is inevitable with any and particularly a novel isotope proxy, especially considering the large interpretational limitations even within the traditional isotopes C and N that have been studied for decades in thousands of publications. In our manuscript we clearly indicate where the discussion shows clear results and where we speculate on controls. We also offer alternatives to interpretations, where necessary. We would argue that this is good scientific practice, anything else would be unacceptable.

We would also argue, that 53 individuals of the same species analysed for three different isotopes (on the same tooth per individual) is not a small sample size. Especially if you consider that a major application of Zn isotopes will be the palaeontological record, where these kind of large sample sizes are rare for a single species. In any case, this sample size is clearly sufficient to document statistically clear differences in Zn isotopes related to, e.g., size of the animal. Again a feature of great interest not only for modern ecological studies but also palaeontology. The size related differences in Zn isotopes holds also without placing individuals into sex categories, which we expand more on now in the discussion and plotted also in Figure 1. The placing into sex categories is of course not always possible for fossils, but in this case, as we had this information it made sense to include it, as it reveals clear differences between large females and large males. Even with the limited sample size for “adult” females, it is clear that these four individuals, whether or not you label them as adult or not, are distinct in their isotopic composition from all other individuals (see also our response to the reviewer comment above 1.2, 1.5 to 1.7 and 1.12).

1.14 Figures – no confidence interval on regression slopes? Can we have the slopes reported?

We have added the confidence intervals and now report the slopes for all figures where relevant.

1.15 Re ethics approval – For my institution I would need to obtain ethics approval for re-purposing samples taken in previous field campaigns - especially if the sampling did not have specific approval for removal of live animals or manipulation of animals. Generally making use of existing biological material is seen as good practice, but not free from ethical responsibility – so I would perhaps refine a statement that ethical approval is not necessary. Some oversight is probably good practice.

We see the reviewers point and must agree. We have thus added the relevant information to the reporting summary.

Reviewer #2 (Remarks to the Author):

2.1 The manuscript is interesting. This is the first time I see the use of the Zinc isotope as a reference in trophic information and even related to the trophic level. This approach and even more so in sharks is quite useful specifically because it allows us to corroborate what is found in the ^{15}N isotopes and relate it to the environmental part in terms of the conditions of the environment where predators tend to feed. I suggested some changes in the title because it refers only to the Zinc isotope, and they work with N and C as well. Additionally, there are some confusing sections that are worth reviewing. I consider that much of the discussion is well addressed although some sessions related to mature females justify that the isotope values could be related to reproductive processes, but at no point do they mention that the mature females had any sign that they were gravid females, so this should be handled with caution. However, I consider that the manuscript is very interesting

and will allow knowing how these new isotopes have a utility in eco-trophic studies specifically in marine predators.

We thank the reviewer for his comments and fair and constructive feedback. We apologies for any confusing sections and have rewritten these and/or added additional information to clarify our arguments and avoid further confusion (see our response to 2.5, 2.6, and 2.7).

Specific Comments

2.2 • Title: Why if the manuscript is focused on trophic ecology using SIA tools of ^{13}C , ^{15}N just mentioned Zinc... I would like to suggest modifying the title... could be: "Shark teeth isotope values document intrapopulation foraging differences related to ontogeny and sex"

We see the reviewers point here and we partially agree. For now, we would, however, like to keep the title as is, as we want to keep the readers focus on the most novel aspect of the study, the inclusion of zinc isotopes to address intrapopulation foraging differences, in line with the suggestions of reviewer 1.

2.3 • Lines 40-41. The ^{13}C does not always increase with the trophic level at least not in top predators such as sharks. Several species had shown that does not exist a positive relationship between ^{15}N and ^{13}C ... is logical to think that ^{15}N must increase with the trophic level is the way to interpret this isotope. But the ^{13}C most of the time showed an opposite trend or any relationship.

We agree with the reviewer. Naturally, $\delta^{13}\text{C}$ is typically not used to infer trophic levels, but rather used to identify the primary producer(s) at the base of the food web. We thought it important to mention here that in some cases a small increase in $\delta^{13}\text{C}$ with trophic level has been observed, but this is of course not always the case, which is why we had placed a ~ in front of the 1 ‰. We agree, however, this might be too little, so we added that $\delta^{13}\text{C}$ may not change at all with trophic level and that $\delta^{13}\text{C}$ values are more commonly applied to identify the primary producer(s) in a food web (lines 51-55).

2.4 • Line 82 Figure 1. The panel d-f represents the data from Males... why only put panels d and e... please check this.

Thank you very much for pointing out this error. We added additional panels to this figure and made sure that all panels are correctly addressed in the caption now.

2.5 • Lines 139-141. How could be possible that the information that you got from the teeth of sharks caught in August of 2012 reflects the diet and habitat condition from October to December 2011? I know that does not exist the turnover rate for this species but you previously mentioned that considering the captivity study from *T. semifasciata* was ~260 days and the incorporation rate of dietary nutrients into dentine collagen is 32 – 83 days. The time that is proposed that reflects these isotopes even is more than 260 days (close to 8.6 months).. Please you need to be more detailed in this approach because is confusing.

We have now expanded our explanation and are more detailed on why our three isotope proxies should represent dietary information from autumn the year prior (see lines 241-248).

2.6 • Lines 170-172. The $\delta^{66}\text{Zn}_{\text{en}}$ start to decrease with the trophic level is logical though the mature individual has the lowest value, so Why you mentioned that the difference in juvenile males could be related to the less variety of prey level? I agreed that juvenile males could consume prey with lower compared with adult males... these lines are a little confusing.

We have rephrased this part (lines 281-284). We agree, it may have been a little confusing. We simply tried to convey that a low $\delta^{66}\text{Zn}_{\text{en}}$ variability in small juveniles may be due to a more restricted diet in terms of prey size and therefore prey trophic levels. In contrast adult males may be able to feed across a range of prey sizes and trophic levels leading to more variable $\delta^{66}\text{Zn}_{\text{en}}$ values.

2.7 • Lines 209-215. The 4 mature females were pregnant?. In the document, I didn't find anything related to this. These lines aboard the importance of this process could affect more the ^{66}Zn in comparison with the other isotopes. but what's the point of this?

Again we see that our phrasing may have led to some confusion. We have rephrased this part (lines 313-332) and have generally shortened the discussion on the adult females, which is now more clearly structured. We do not know whether these females were pregnant at the time of tooth formation. However, since these four individuals are so distinct in terms of their isotopic composition, we felt it necessary to address a potential physiological factor that could cause differences in isotopic fractionation factors for these individuals only, as they are the only mature female individuals. However, even if they were pregnant at the time of tooth formation, pregnancy influencing the animal's metabolism cannot explain the observed isotopic variability of these four individuals, as we discuss in the manuscript (lines 317-325). Although pregnancy may indirectly affect the isotope values, due to different habitat use and or consumption of different prey to satisfy their different physiological demands. In any case, these individuals were likely foraging in a different habitat, in line with literature on habitat use of this population, and likely feeding on higher trophic level prey than juveniles and generally adult males based on our isotope results.

References

- Bourgon, N. *et al.* Diet of a Late Pleistocene early modern human from Southeast Asia inferred from zinc and carbon isotopes. *J. Hum. Evol.* **161**, 103075 (2021).
- Bourgon, N. *et al.* Zinc isotopes in Late Pleistocene fossil teeth from a Southeast Asian cave setting preserve paleodietary information. *Proc. Natl. Acad. Sci.* **117**, 4675–4681 (2020).
- Horner, T. J., *et al.* Bioactive trace metals and their isotopes as paleoproductivity proxies: An assessment using GEOTRACES-era data. *Global Biogeochemical Cycles*, **35**, e2020GB006814. (2021).
- Jaouen, K., Beasley, M., Schoeninger, M., Hublin, J. J. & Richards, M. P. Zinc isotope ratios of bones and teeth as new dietary indicators: results from a modern food web (Koobi Fora, Kenya). *Sci. Rep.* **6**, 26281 (2016).
- Jaouen, K., *et al.* A Neandertal dietary conundrum: Insights provided by tooth enamel Zn isotopes from Gabasa, Spain. *Proceedings of the National Academy of Sciences*, **119**, e2109315119. (2022).
- Little, S. H., Vance, D., Walker-Brown, C. & Landing, W. M. The oceanic mass balance of copper and zinc isotopes, investigated by analysis of their inputs, and outputs to ferromanganese oxide sediments. *Geochim. Cosmochim. Acta* **125**, 673–693 (2014).

McCormack, J. *et al.* Trophic position of *Otodus megalodon* and great white sharks through time revealed by zinc isotopes. *Nat. Commun.* **13**, 2980 (2022).

McCormack, J. *et al.* Zinc isotopes from archaeological bones provide reliable trophic level information for marine mammals. *Commun. Biol.* **4**, 683 (2021).

Rossman, S., Ostrom, P. H., Gordon, F., & Zipkin, E. F. Beyond carbon and nitrogen: guidelines for estimating three-dimensional isotopic niche space. *Ecology and Evolution*, **6**, 2405-2413. (2016).